# Comparison of Early and Late Intubation in COVID-19 and Its Effect on Mortality

**DOI:** 10.3390/ijerph19053075

**Published:** 2022-03-05

**Authors:** Benjamin McKay, Matthew Meyers, Leah Rivard, Holly Stankewicz, Jill C. Stoltzfus, Guhan Rammohan

**Affiliations:** 1Temple/St. Luke’s Medical School, St. Luke’s University Hospital, 801 Ostrum Street, Bethlehem, PA 18015, USA; benjamin.mckay@sluhn.org; 2Department of Emergency Medicine, St. Luke’s University Health Network, 801 Ostrum Street, Bethlehem, PA 18015, USA; matthew.meyers@sluhn.org (M.M.); leah.rivard@sluhn.org (L.R.); holly.stankewicz@sluhn.org (H.S.); jill.stoltzfus@sluhn.org (J.C.S.)

**Keywords:** COVID-19, intubation, emergency department, qSOFA

## Abstract

Background: Best practices for management of COVID-19 patients with acute respiratory failure continue to evolve. Initial debate existed over whether patients should be intubated in the emergency department or trialed on noninvasive methods prior to intubation outside the emergency department. Objectives: To determine whether emergency department intubations in COVID-19 affect mortality. Methods: We conducted a retrospective observational chart review of patients who had a confirmed positive COVID-19 test and required endotracheal intubation during their hospital course between 1 March 2020 and 1 June 2020. Patients were divided into two groups based on location of intubation: early intubation in the emergency department or late intubation performed outside the emergency department. Clinical and demographic information was collected including comorbid medical conditions, qSOFA score, and patient mortality. Results: Of the 131 COVID-19-positive patients requiring intubation, 30 (22.9%) patients were intubated in the emergency department. No statistically significant difference existed in age, gender, ethnicity, or smoking status between the two groups at baseline. Patients in the early intubation cohort had a greater number of existing comorbidities (2.5, *p* = 0.06) and a higher median qSOFA score (3, *p* ≤ 0.001). Patients managed with early intubation had a statistically significant higher mortality rate (19/30, 63.3%) compared to the late intubation group (42/101, 41.6%). Conclusion: COVID-19 patients intubated in the emergency department had a higher qSOFA score and a greater number of pre-existing comorbidities. All-cause mortality in COVID-19 was greater in patients intubated in the emergency department compared to patients intubated outside the emergency department.

## 1. Introduction

Over the span of a few months in early 2020, the coronavirus disease 2019 (COVID-19) pandemic spread from an isolated outbreak in the city of Wuhan, China to a global catastrophe of unrivaled proportions during this century [1,2,3]. As the novel virus spread globally, the medical community was forced to manage critically ill patients without data-based evidence for best practices. Many of the early treatment protocols for SARS-CoV-2 were devised from trial and error, theoretical calculations, or from extrapolating from previously successful therapies for treating similar pathologies [4]. The understanding of the pathophysiology and management of critically ill COVID-19 patients continues to evolve. Previously reported data from the initial outbreak in Wuhan, China demonstrate that older patients (age > 65 years old), patients with underlying medical conditions, and patients who develop acute respiratory distress syndrome (ARDS) have higher mortality from SARS-CoV-2 pneumonia [5]. For critically ill patients who progress to COVID-19-related respiratory failure, the timing and threshold for escalating from noninvasive oxygen supplementation to mechanical ventilation is unclear.

The emergency department management of COVID-19 patients presents a unique dilemma for clinicians. In particular, the decision to perform endotracheal intubation or utilize noninvasive forms of oxygenation is a point of debate. The urge to promptly improve oxygenation and the work of breathing is weighed against complications such as ventilator-associated pneumonia, ventilator-induced lung injury, hemodynamic changes, and complications related to prolonged sedation and immobilization which are inherent to intubated patients [6]. The risk of peri-intubation hypoxemia (S_p_O_2_ < 80%), previously reported as 10% of intubations, is thought to be greater in COVID-19 patients [7].

The aim of our study is to expand on these data comparing patients managed with endotracheal intubation for COVID-19-related respiratory failure. To the best of our knowledge, this is the first study comparing outcomes in COVID-19 patients intubated in the emergency department and in the intensive care unit. We hypothesize that, counter to current practice guidelines recommending early endotracheal intubation of critically ill COVID-19 patients, no mortality benefit exists in patients who are intubated emergently in the emergency department versus patients who are intubated in an intensive care unit (ICU), operating room, or in other more controlled settings.

## 2. Materials and Methods

This was a retrospective observational chart review study without any patient interventions. Data were collected from a large healthcare network comprising 12 hospitals in eastern Pennsylvania and western New Jersey. Due to the nature of the study, informed consent was waived. Data were provided by the study hospital via the electronic medical record (EMR). Patients were identified via the EMR as having a confirmed positive COVID-19 test result that was ordered between 1 March 2020 and 1 June 2020. Furthermore, from this list, all patients who had received endotracheal intubation were identified and extracted to their own data set, and the location and timing of their intubation was determined via manual chart review. Demographic information and clinical outcomes were documented and recorded, also via chart review. In order to compare the populations of the two groups, we collected past medical conditions as presented in the initial-encounter H&P to look for the comorbidities of congestive heart failure, diabetes, hypertension, hyperlipidemia, malignancy, immunocompromised state, CAD, COPD, asthma, and CKD, and assigned each comorbidity present on admission 1 point to give a total comorbidity score.

Inclusion criteria were adult patients who had a confirmed positive COVID-19 polymerase chain reaction (PCR) test and went on to require endotracheal intubation within the hospital network between 1 March 2020 and 1 June 2020. Exclusion criteria were patients under the age of 18, patients that did not require endotracheal intubation, and patients who may have had symptoms where COVID-19 was suspected but who had a negative COVID-19 PCR test result. We defined early intubation (EI) as patients who were intubated in the ED and late intubation (LI) as patients who were intubated in a more controlled setting such as the ICU or operating room. There was no specific time which defined EI vs. LI, rather patients were stratified based on the disposition and location of intubation.

Results were analyzed by conducting separate unadjusted comparisons between intubated and non-intubated ED patients using independent samples tests or Mann–Whitney rank sum tests for our continuous outcomes, as appropriate, and chi-square tests for our categorical outcomes. We analyzed data using SPSS version 25 (IBM Corp., Armonk, NY, USA), with *p* < 0.05 denoting statistical significance and no adjustment for the multiple comparisons.

This study was approved by the IRB and was found to be in accordance with the tenets of the Declaration of Helsinki.

## 3. Results

A total of 131 COVID-19-positive patients requiring intubation during their hospitalization were analyzed. Thirty (22.9%) of these patients were intubated in the ED and were designated as early intubation (EI), while the remaining 101 (77.1%) were intubated elsewhere in the hospital and were designated as late intubation (LI). Overall, these two cohorts had similar baseline demographic characteristics (Table 1). The mean age for those intubated in the ED was 68.5 + 12.1, and it was 64.4 + 13.6 for patients intubated elsewhere (*p* = 0.12). In total, 33.3% of the EI group and 35.6% of the LI group were female (*p* = 0.82), and 41.4% and 54.1% of the EI and LI groups, respectively, were non-white individuals (*p* = 0.23). Fifty percent of the EI and 56.4% of the LI cohort were identified as never having smoked, with the remainder of patients either being current smokers or past smokers (*p* = 0.53). Lastly, both groups had similar prior existing comorbidities, with a median of 2.5 and 2 in the EI and LI groups, respectively, (*p* = 0.06).

Patients intubated in the ED had a higher mortality rate during hospitalization, in contrast to those intubated elsewhere in the hospital and later during their admission after leaving the emergency department (Table 2). The death rate in the EI group was 63.3%, in contrast to 41.6% in the LI group, which is statistically significant (*p* = 0.04). However, it should be noted that there was a statistically significant difference in the baseline quick sequential organ failure assessment (qSOFA) scores between the cohorts, with ED intubations having a higher median qSOFA score (*p* < 0.0001). Patients in the EI cohort had a median qSOFA score of 2.53 and patients in the LI cohort had a median qSOFA score of 1.43.

## 4. Discussion

We hypothesized that no mortality benefit existed for patients that were intubated emergently in the emergency department (EI) versus patients who were intubated later during their hospital course (LI). The data demonstrated that patients who were intubated emergently in the emergency department in fact had a higher in-hospital mortality rate (63.3% versus 41.6%) than those who were intubated later in their admission.

While the overall demographics of age, gender, smoking status, and race did not show significant differences between the two groups, there was a large, though not statistically significant, difference between the total number of comorbidities (2.6 versus 2.0, *p* = 0.06). The EI patients also had a higher initial acuity based on their qSOFA scores (3 versus 1). Prior to the COVID-19 pandemic, it was believed that early intubation for patients presenting with hypoxic respiratory failure provided the best patient outcomes.

Reflecting on several months of managing critically ill COVID-19 patients, the Chinese Society of Anesthesiology Task Force provided recommendations in February 2020 for criteria for intubating patients with respiratory failure due to COVID-19 [8]. These criteria include critically ill patients who, after two hours of noninvasive oxygen supplementation, remain hypoxemic, remain in respiratory distress, or have unresolved tachypnea. This recommendation was based on the theory that early intubation may be physiologically protective by reducing a phenomenon known as self-induced lung injury [9]. Expert opinion articles comment that increased respiratory effort may lead to self-induced lung injury (SILI). It is thought that intubation and mechanical ventilation have protective effects by diminishing inspiratory effort and tidal volumes, thus limiting the effect of SILI [9,10]. Some of the critical care providers who initially cared for COVID-19 patients in Wuhan, China lamented that patients developed an oxygen debt and had been intubated too late in their disease course [11]. This led to an early hypothesis-driven advisory that COVID-19 patients should be ventilated early in the disease process to prevent lung injury.

However, new data regarding COVID-19 management and outcomes have called this paradigm into question [6,12,13,14,15]. The pathophysiology of COVID-19 differs from more traditional acute respiratory distress illness, and it is more likely to respond to noninvasive forms of oxygenation (e.g. high-flow nasal cannula) [12]. As such, early management of COVID-19-induced hypoxemia revolves around noninvasive forms of oxygenation [13]. Some authors caution against performing early intubation in COVID-19 patients by citing a lack of clinical evidence for this practice, and they note that the proposed hypothesis of COVID-19-patient-induced SILI is more theoretical [6,14]. Literature studies regarding the timing of intubation of COVID-19 patients are limited, though small studies have demonstrated conflicting mortality rates in patients intubated earlier in their clinical course [15,16].

It should be noted that some literature studies recommended against utilizing noninvasive ventilation for fear this would aerosolize COVID-19 particles [17]. It is possible that some emergency department providers bypassed noninvasive oxygenation and performed intubation for this reason.

We believe our data are in support of delayed intubation in cases of COVID-19. There has been conflicting evidence regarding timing of intubation, and this has evolved as more has become known about this disease. As our data show, the sicker patients, as evidenced by higher qSOFA scores, were intubated earlier; however, this did increase mortality. This is likely to be due to the increased lung injury that occurred due to mechanical ventilation. We believe that further studies would elucidate the reasons why other strategies in severe COVID-19 would be more beneficial in improving mortality than early intubation. These would include high-flow nasal cannula, noninvasive ventilation, proning, and other therapeutic interventions such as glucocorticoids. Early intubation, as mentioned, is likely to be causing more harm due to worsening lung injury earlier in the disease progression, thus causing worsening hypoxemia and increased multi-organ failure and leading to a higher mortality. We believe our data confirm this hypothesis.

As understanding of COVID-19 pathophysiology evolved, many authors noted differences in COVID-19 lung injury compared to acute respiratory distress syndrome from other etiologies [18]. Typical ARDS is associated with decreased lung compliance, which is managed with lung-protective ventilation strategies [19]. However, mechanical ventilation in COVID-19 patients is associated with a high mortality rate and may actually worsen acute lung injury [20,21].

Today, the interventions recommended for preventing respiratory failure in COVID-19 are centered around delaying endotracheal intubation and utilizing noninvasive modes of oxygenation [22]. These modes include mid-flow nasal cannula, high-flow nasal cannula, CPAP or BIPAP, and self-proning strategies [23,24,25,26]. At present, endotracheal intubation is viewed as a last resort for refractory hypoxia. Providers should tolerate lower oxygen saturations so long as patients do not exhibit changes in mental status or respiratory fatigue. The data presented here add to the growing body of literature supporting delayed intubation for COVID-19 patients.

## 5. Limitations

This study does have several limitations. The first is that the decision by providers to utilize noninvasive oxygenation versus performing endotracheal intubation was left to provider discretion. No standard criteria were utilized to determine the need for endotracheal intubation. Secondly, this collection of data was obtained early in the COVID-19 pandemic. Since this time, many adjuvant therapies and recommendations have been published which reflect more comfort with noninvasive modes of oxygenation before performing intubation. In addition, noninvasive oxygenation technology has evolved to minimize viral particle aerosolization. The third limitation is that a selection bias may exist. Patients who were intubated earlier in their hospital course were likely to be sicker at presentation, and thus would be expected to have a higher mortality rate. This is evidenced by the higher qSOFA score in the EI cohort versus the LI cohort in this study. Also, the qSOFA score has been proven to be of little utility in the evaluation of patients with COVID-19 since the septic shock occurs in a late phase of the disease. We could have used other well known prediction scores specific to COVID, but these were developed after our study was conducted. We also thought to run a Propensity Score Matched analysis on the sample, to evaluate if the difference in mortality still exists despite the difference in qSOFA score, however, our sample size was too small to run this analysis. The retrospective nature and small sample size of this study are also added limitations. Future research would benefit from a large, prospective trial comparing EI versus LI.

Despite the limitations mentioned above, these data add to the current and developing literature suggesting that no mortality benefit exists with earlier intubation of SARS-CoV-2 patients. In fact, these data suggest that early intubation may be detrimental.

## 6. Conclusions

In reviewing 131 COVID-19 patients intubated between 1 March 2020 and 1 June 2020, we found that patients intubated in the emergency department had a higher qSOFA score and a greater number of pre-existing comorbidities than patients intubated outside the emergency department. Patients intubated in the emergency department had an in-hospital mortality rate of 63.3%, compared to 41.6% for patients intubated outside the emergency department.

## 7. Article Summary

1.Why is this topic important?Determining the timing of intubation in COVID-19 hypoxic respiratory failure is a topic of debate. Studies are conflicting regarding the optimal timing of intubation in COVID-19, with regard to its effect on lung injury and mortality.

2.What does this study attempt to show?Early intubation in COVID-19 patients with hypoxic respiratory failure may be detrimental.

3.What are the key findings?Patients with higher qSOFA scores and hypoxic respiratory failure secondary to COVID-19 have higher mortality when intubated earlier, in the emergency department.

4.How is patient care impacted?Based on the findings, we would recommend delaying intubation in COVID-19 hypoxic respiratory failure in favor of other noninvasive modes of oxygenation.

## Figures and Tables

**Table 1 ijerph-19-03075-t001:** Demographics of patients with COVID-19 intubated early vs. late.

		Intubated in ED (EI) (n = 30)	Not Intubated in ED (LI) (n = 101)	*p*-Value
Age (mean + SD)		68.5 + 12.1	64.4 + 13.6	0.12
Gender (n, %)	Female	10 (33.3%)	36 (35.6%)	0.82
	Male	20 (66.7%)	65 (64.4%)	
Race (n, %)	White	17 (58.6%)	45 (45.9%)	0.23
	Non-White	12 (41.4%)	53 (54.1%)	
Smoking Status (n, %)	Prior or current smoker	15 (50%)	44 (43.6%)	0.53
	Never smoked	15 (50%)	57 (56.4%)	
Total Comorbidities (median, range)		2.5 (0–5)	2 (0–6)	0.06

EI = early intubation in emergency department; LI = intubation outside emergency department.

**Table 2 ijerph-19-03075-t002:** Mortality and qSOFA score in early vs. late intubation patients with COVID-19.

	Intubated in ED (EI) (n = 30)	Not Intubated in ED (LI) (n = 101)	*p*-Value
Expired (n, %)	19 (63.3%)	42 (41.6%)	0.04
qSOFA (median, range)	3 (1-3)	1 (0-3)	< 0.0001

EI = early intubation in emergency department; LI = intubation outside emergency department; qSOFA = quick sequential organ failure assessment.

## Data Availability

Data available on request due to restrictions of privacy implemented by the institution. The data presented in this study are available on request from the corresponding author. The data are not publicly available due to institutional restrictions.

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
