# Peer review of "Comparison of Early and Late Intubation in COVID-19 and Its Effect on Mortality"

_ijerph, 2022, doi:10.3390/ijerph19053075_

Round 1

Reviewer 1 Report

I read the paper "Comparison of Early and Late Intubation in COVID-19 and its Effect on Mortality" with great pleasure. The authors suggested, that patients intubated in the emergency department had an in-hospital mortality rate of 63.3% compared to patients intubated outside of the emergency department, who had in-hospital mortality of 41.6%. This is very interesting information. Undoubtedly, this topic requires further research. However, in my opinion, it is a very well-written research paper in all respects. Therefore, without a doubt, I recommend this work for publication.

Author Response

Thank you for your support of this paper.  It appears, there are no other revisions to be made based on your comments.  Thank you.

Reviewer 2 Report

The authors addressed well the comments. However, the discussion could be improved by discussing the authors findings, not only reviewing the literature.

Author Response

I have attached the modified discussion, with the addition bolded and italicized.  Thank you for the feedback.  Hopefully, this addressed the editors concerns.  Thank you.

Round 2

Reviewer 2 Report

The authors addressed well the comments.